# Statins and Bempedoic Acid: Different Actions of Cholesterol Inhibitors on Macrophage Activation

**DOI:** 10.3390/ijms222212480

**Published:** 2021-11-19

**Authors:** Rebecca Linnenberger, Jessica Hoppstädter, Selina Wrublewsky, Emmanuel Ampofo, Alexandra K. Kiemer

**Affiliations:** 1Department of Pharmacy, Pharmaceutical Biology, Saarland University, Campus C2.3, 66123 Saarbrücken, Germany; rebecca.linnenberger@uni-saarland.de (R.L.); j.hoppstaedter@mx.uni-saarland.de (J.H.); 2Institute of Clinical and Experimental Surgery, Saarland University, 66424 Homburg, Germany; selina.wrublewsky@uks.eu (S.W.); emmanuel.ampofo@uks.eu (E.A.)

**Keywords:** natural compounds, polarization, bone marrow-derived macrophages, LPS, phagocytosis, arginase, HMG-CoA reductase, inflammasome, GILZ, KLF2

## Abstract

Statins represent the most prescribed class of drugs for the treatment of hypercholesterolemia. Effects that go beyond lipid-lowering actions have been suggested to contribute to their beneficial pharmacological properties. Whether and how statins act on macrophages has been a matter of debate. In the present study, we aimed at characterizing the impact of statins on macrophage polarization and comparing these to the effects of bempedoic acid, a recently registered drug for the treatment of hypercholesterolemia, which has been suggested to have a similar beneficial profile but fewer side effects. Treatment of primary murine macrophages with two different statins, i.e., simvastatin and cerivastatin, impaired phagocytotic activity and, concurrently, enhanced pro-inflammatory responses upon short-term lipopolysaccharide challenge, as characterized by an induction of tumor necrosis factor (TNF), interleukin (IL) 1β, and IL6. In contrast, no differences were observed under long-term inflammatory (M1) or anti-inflammatory (M2) conditions, and neither inducible NO synthase (iNOS) expression nor nitric oxide production was altered. Statin treatment led to extracellular-signal regulated kinase (ERK) activation, and the pro-inflammatory statin effects were abolished by ERK inhibition. Bempedoic acid only had a negligible impact on macrophage responses when compared with statins. Taken together, our data point toward an immunomodulatory effect of statins on macrophage polarization, which is absent upon bempedoic acid treatment.

## 1. Introduction

Mevastatin, exhibiting potent hypocholesterolemic activity, was isolated in 1976 from the fungus *Penicillium citrinum*. Shortly thereafter, two other statins, namely pravastatin and lovastatin, were discovered [1]. Whereas the semi-synthetic simvastatin solely differs in an alkyl moiety from lovastatin, synthetic statins, such as cerivastatin and atorvastatin, only have the pharmacophore in common with their ancestors [2]. Today, statins are the first-line treatment of cardiovascular diseases (CVDs), which represent the leading cause of death worldwide [3].

Due to hypercholesterolemia being one of the underlying conditions of atherosclerotic diseases, statin prescriptions have been rising within the last years, and statins are today the most prescribed class of drugs worldwide [4]. From natural compound-derived statins to new synthetic ones, the mode of action has remained the same: This class of drugs interferes with the rate-limiting step of cholesterol synthesis in the liver by competitively inhibiting hydroxy-methyl-glutaryl-coenzyme A (HMG-CoA) reductase, resulting in enhanced low-density lipoprotein (LDL) clearance from the circulation [5,6,7]. Besides their lipid-lowering actions, statins exert pleiotropic effects, which might be due to the impaired synthesis of isoprenoids as intermediates of the mevalonate pathway. The reduced synthesis of prenylated proteins, such as Ras and Rho family small GTPases, results in altered cell signaling [8].

Clinical studies on atherosclerosis have shown that these pleiotropic effects are beneficial and linked them to antioxidant or anti-inflammatory effects and plaque stabilization [9,10]. Therefore, statins have been suggested for the treatment of respiratory conditions such as pneumonia and acute respiratory distress syndrome [11,12]. They have also been reported to be beneficial in different bacterial infections [13,14], and their benefits in the treatment of COVID-19 are currently under investigation [15]. Consistent clinical evidence for the benefits of statin use in a broader range of inflammatory conditions is lacking, which is why statins are still only approved for primary and secondary prevention of cardiovascular events.

It is, however, very much in doubt whether the statin-mediated inhibition of inflammation in the context of atherosclerosis is independent of its cholesterol-lowering actions since recently published studies suggested that statins may activate inflammatory pathways, and non-statin cholesterol-lowering drugs also have anti-inflammatory potential [16,17,18,19,20,21,22,23,24].

Macrophages have been postulated as one target cell type of statin actions. They represent a heterogeneous cell population that is characterized by high plasticity. Both exogenous and endogenous factors determine macrophage polarization, including, but not limited to, pathogen-associated molecular patterns, such as lipopolysaccharide (LPS), danger-associated molecular patterns, but also natural compounds [25,26,27]. They mediate their response towards pathogen- or danger-associated molecular patterns by the release of cytokines and small molecules.

Reports on whether and how statins interact with macrophages are conflicting, however. Both pro- and anti-inflammatory effects, which result from altered isoprenylation and activation of stress kinases, such as c-Jun N-terminal kinase (JNK) or extracellular signal-regulated kinase (ERK), have been described. In addition, statins have been suggested to affect the NOD- (nucleotide-binding oligomerization), LRR- (leucine-rich repeat), and pyrin domain-containing protein 3 (NLRP3) inflammasome and induce the expression of the transcription factor Krüppel-like factor 2 (KLF2), a potent inhibitor of metabolic inflammation [16,28,29,30,31,32].

The novel cholesterol-lowering agent bempedoic acid (ETC-1002) was recently approved for the treatment of hypercholesterolemia. This small synthetic prodrug is suggested to act only in hepatocytes. The compound requires a hepatocyte-specific enzyme, the very long-chain acyl-CoA synthetase-1 (gene name: *Slc27a2*), for transformation into its active form, ETC-1002-CoA, which then inhibits ATP-citrate lyase (gene name: *Acly*) upstream of HMG-CoA reductase [33,34]. Clinical studies revealed a reduction of LDL-cholesterol and total cholesterol, but not triglycerides, upon bempedoic acid administration [35].

The pharmacokinetic properties of bempedoic acid aim to prevent muscle-related side effects, which frequently occur under statin treatment. These adverse effects of statins have been associated with impaired mitochondrial function and disturbed calcium homeostasis within muscle cells [36]. Our previously published data also suggest an involvement of the glucocorticoid-induced leucine zipper (GILZ, gene name *Tsc22d3*), a protein formerly mainly known for its anti-inflammatory properties in leukocytes: GILZ was induced by statins in muscle cells and contributed to statin-mediated myotoxic and anti-myogenic effects [37].

In addition to its cholesterol-lowering activity, bempedoic acid treatment decreases high-sensitivity C-reactive protein serum levels, suggesting that the compound exhibits anti-inflammatory properties [35].

As both statins and bempedoic acid show anti-inflammatory activity in vivo, we hypothesized that both types of cholesterol-lowering agents might affect macrophage responses. The aim of the present study was to systematically test different aspects of macrophage activation upon statin and bempedoic acid treatment.

## 2. Results

### 2.1. Modulation of Inflammatory and Anti-Inflammatory Mediator Expression in Statin-Treated Macrophages

Potential effects of two statins, i.e., simvastatin (Sim, 2 µM) and cerivastatin (Cer, 0.5 µM), on macrophages were investigated at non-toxic concentrations (Appendix A). These relatively high statin concentrations are in accordance with the literature [37,38] and were based on our observations of higher *Hmgcr* and *Acly* levels after statin treatment, as well as the known negative feedback regulation of HMG-CoA reductase in cell culture (Appendix A) [39].

Nuclear factor ‘kappa-light-chain-enhancer’ of activated B-cells (NF-κB) and activator protein 1 (AP-1) are the main transcription factors involved in macrophage inflammatory activation. Statin treatment of a macrophage reporter cell line activated NF-κB/AP-1, and an additional short-term LPS challenge (4 h LPS) resulted in an even higher amplitude of inflammation (Figure 1A).

Primary bone marrow-derived macrophages (BMMs) were either treated with statins alone or in combination with different stimuli, namely a short-term inflammatory activation (4 h LPS), an M1 (24 h LPS/interferon-gamma (IFNγ)), or an M2 (24 h interleukin (IL) 4) treatment scheme. We then quantified the cytokines TNF and IL1β, NO, and M1/M2-associated gene expression levels. Both statins amplified the LPS-induced production of TNF and IL1β (Figure 1C,E). Moreover, statin treatment increased the levels of *Il6* mRNA in LPS-treated cells (Figure 1F). *Il1b* and *Il6* mRNA levels were above background levels upon treatment with either statin in the absence of LPS, while *Tnf* mRNA was decreased (Figure 1B,D,F, Appendix A). No modulatory effect of statins was observed under M1 conditions (Figure 1B,D,F). Statins did not influence NO release under inflammatory conditions, and only cerivastatin slightly affected *Nos2* mRNA (Figure 1G–H, Appendix A).

Next, we sought to examine whether statins exert modulatory effects on markers associated with anti-inflammatory actions. Statins significantly induced arginase-1 and GILZ (gene names: *Arg1*, *Tsc22d3*) under every tested condition (Figure 2A,B, Appendix A). While *Il10* and *Tgfb1* expression were hardly affected by statin treatment, *Mrc1* levels were decreased (Figure 2C–E, Appendix A). The transcription factor *Klf2* was induced by statins (Figure 2F, Appendix A). Its induction followed a similar pattern as observed for *Tsc22d3*, suggesting crosstalk between these anti-inflammatory mediators. This assumption was further supported by the observation that statin-induced *Klf2* expression was reduced in *Tsc22d3* knockout macrophages (Appendix A).

In summary, these data point towards a unique statin-induced modulation of the macrophage phenotype regarding expression levels of M1/M2 markers after statin treatment. While the induction of M1-associated genes points towards a pro-inflammatory phenotype, an exacerbation of inflammatory responses may be limited by the selective induction of anti-inflammatory mediators.

### 2.2. Statins Modulate the Phagocytotic Activity of Macrophages

Phagocytosis represents one of the major macrophage functions, which has been shown to be altered in atherosclerosis [40]. Our data show a distinct reduction of phagocytotic activity by both simvastatin and cerivastatin when added to untreated cells or combined with short-term inflammatory activation (Figure 3A,B,G: 12 h time point; Appendix A: additional images). Both M1 and M2 polarization reduced the phagocytotic activity in a time-dependent manner (Figure 3C), as reported previously [41,42]. The presence of statins during polarization showed no additional effect (Figure 3D–F). 

### 2.3. ERK Activation Contributes to Statin-Induced Inflammation

We then sought to elucidate the molecular mechanisms underlying statin-mediated modulation of the macrophage phenotype. It has been reported that macrophages produce cholesterol even in serum-containing media and that cholesterol biosynthesis can be reduced by statin treatment [43]. Macrophages can, however, quickly replenish cholesterol from the media if their cholesterol levels decline [44]. Thus, the total cellular cholesterol content is not necessarily affected when de novo synthesis is shut down. This reflects the in vivo situation, in which cholesterol can be scavenged from the surrounding environment, e.g., from the bloodstream, which is why we performed all assays in the presence of serum. To determine whether the cholesterol content indeed remains unchanged after statin treatment, we measured cellular cholesterol levels and found that they were not affected by statins (Figure 4A). Interestingly, we observed that the cellular cholesterol content was reduced by statins under low-serum conditions, i.e., under conditions that restrict cholesterol replenishment from the media (Appendix A).

Since statins’ pleiotropic effects have been described to be related to attenuated protein prenylation, we co-treated cells with mevalonate (MVA) as an intermediate of the mevalonate pathway. As shown in Figure 4B, the statin-mediated enhancement of LPS-induced NF-κB/AP-1 activity was abolished by MVA priming during cerivastatin treatment, but not during simvastatin treatment, which may point towards distinct mechanisms or different kinetics. A similar effect was observed when farnesyl pyrophosphate (FPP) or geranylgeranyl pyrophosphate (GGPP) were added instead of MVA, suggesting that isoprenylation is required to induce cerivastatin-mediated downstream effects (Appendix A). Interestingly, cerivastatin has been shown to reduce the expression of genes involved in the MVA pathway to a higher degree than simvastatin [45], implying that this pathway may be especially vulnerable to cerivastatin-mediated interference.

Statins have been shown to activate the NLRP3 inflammasome in BMMs [46]. To determine a potential involvement of the NLRP3 inflammasome in the statin-induced increase of IL1β secretion, we used *Nlrp3* KO BMMs. The viability of statin-treated cells was not affected by *Nlrp3* knockout (Appendix A). As shown in Figure 4C, IL1β release was NLRP3 inflammasome-independent.

Gene expression data suggested an upregulation of *Tlr4*, which may contribute to the enhanced susceptibility to LPS of statin-treated cells. In contrast, the expression of neither *Nlrp3* nor *MyD88* or *Tlr2* was altered in the presence of statins (Figure 4D).

Since ERK represents an essential regulator in inflammatory macrophage activation [26,47], we hypothesized that ERK is involved in statin-mediated effects in macrophages. Statin treatment led to an enhanced ERK activation (Figure 4E,F). The addition of the MEK/ERK inhibitor PD98059 prior to statin treatment revealed that statin-induced activation in short-term LPS-treated cells was entirely abolished by the inhibitor (Figure 4G), suggesting ERK as a mediator of statin-facilitated inflammatory macrophage activation.

### 2.4. Bempedoic Acid Treatment Has Minimal Impact on the Phenotype of Macrophages

We then characterized the effect of bempedoic acid on macrophages at the highest concentration at which the solvent showed no effect and at which the compound was neither toxic in BMMs nor in RAW 264.7 cells (25 µM, Appendix A).

The enzyme that converts the prodrug to its active form, i.e., *Slc27a2*, was barely expressed in BMMs, suggesting that potential effects of bempedoic acid on macrophages would be caused by the prodrug (Appendix A). This assumption was further supported by the observation that different types of human macrophages, i.e., monocyte-derived and alveolar macrophages, also expressed very low levels of *SLC27A2* (Appendix A).

In contrast to the pronounced effects of statins on macrophages, bempedoic acid treatment affected neither NF-κB/AP-1 activity nor cytokine transcript levels, IL1β secretion, or ERK activity (Figure 5A,B,D–F, Appendix A). Only a minor increase of TNF protein levels was detectable in LPS-activated cells (Figure 5C).

The phagocytotic activity was not altered by bempedoic acid when cells were otherwise left untreated (Figure 5G). However, bempedoic acid was able to partially rescue the M2-associated decline in phagocytotic activity, while no effect of bempedoic acid was observed in cells under short-term inflammatory or M1 conditions (Figure 5H, Appendix A).

## 3. Discussion

For decades, statins have been the gold standard for the treatment of CVD [48]. They have been suggested to exert pleiotropic effects beyond their cholesterol-lowering actions [2]. Previous publications reported pro- and anti-inflammatory responses in macrophages employing different models and treatment schemes [16,17,22,29,49,50,51]. Thus, we sought to investigate the influence of statins on macrophages during polarization and in short-term inflammation.

Our data showed that statins skew macrophages towards a unique mixed phenotype, both under otherwise unstimulated conditions and after short-term inflammatory activation, but not during M1/M2 polarization. The statin effects were characterized by enhanced inflammatory cytokine production but unaltered NO release. The enhanced cytokine release is in accordance with previously published studies investigating the influence of statins on inflammatory responses [17,50,52,53,54]. However, these studies differ regarding the investigated cell type, the treatment scheme, or the respective inflammatory stimulus. Kuijk et al. used a human monocytic cell line [52], Hohensinner et al. investigated atorvastatin effects [53], and Matsumoto et al. showed elevated TNF levels in RAW 264.7 cells [50]. Kiener et al. used a setup that was most similar to our short-term inflammation model, although human peripheral monocytes were used. In accordance with our results, their study showed elevated LPS-induced cytokine production in cells pre-treated with statins [17].

While we observed a distinct effect of statins on untreated or LPS-activated macrophages, statins did not affect transcript levels of *Tnf*, *Il1b*, and *Il6* when present during M1/M2 polarization. The study by Hohensinner et al. points toward the same direction, showing an unaltered macrophage polarization program after atorvastatin treatment in human macrophages, as suggested by unaltered levels of CD80, CD206, IL6, and IL10 [53].

Despite the induction of inflammatory cytokines, our findings do not show a clear inflammatory phenotype since we observed an unaltered NO release upon statin treatment. This observation might be linked to an increased expression of arginase, which represents a classical marker of M2 macrophages as it antagonizes NO production.

In fact, several studies showed anti-inflammatory effects of statins on monocyte and macrophage inflammatory responses, such as decreased iNOS in a macrophage cell line [38] and CRP-induced chemokine secretion in human monocytes [55], as well as decreased TNF levels in human monocytes ex vivo [56]. The latter can be explained by a 24 h LPS treatment scheme, which might have led to a relatively anti-inflammatory state termed endotoxin tolerance and therefore strongly differs from our experimental setup [55]. The lack of characterization of macrophage phenotypes, species-specific differences, or chemical properties of the used statins might also contribute to the differences to our results. Of note, many in vitro studies were conducted in serum-free media. This approach does not reflect physiologic conditions and may also heavily influence the macrophage phenotype [20,29,57].

There are discrepancies regarding the proposed mechanisms by which statins have been suggested to induce a pro- or anti-inflammatory response. On the one hand, physicochemical interactions between statins and cell membranes have been described, resulting in impaired lipid rafts and membrane fluidity [58]. On the other hand, interactions between statins and cellular signaling pathways, either cholesterol-dependent or isoprenoid-dependent, have been suggested to cause downstream effects [59].

Our data demonstrate a clear activation of NF-κB/AP-1, which represent central inflammatory transcription factors. This finding is in accordance with Healy et al., who showed an increase of NF-κB activity after atorvastatin treatment in BMMs [22]. Interestingly, the study also showed that statins disrupted the complex between the small GTPAse Rac1 and its negative regulator, the Rho guanine nucleotide dissociation inhibitor (RhoGDI), an interaction that is dependent on protein isoprenylation. This led to increased active Rac1 levels in monocytes and macrophages, which may be the cause of some of the pro-inflammatory statin effects in macrophages, including enhanced IL1β secretion.

Similarly, Akula et al. showed that simvastatin facilitated IL1β maturation and secretion in response to LPS [60]. This effect was reversible by GGPP addition, which restored GGTase-I-mediated prenylation of Rac1. Thus, the authors speculated that the pro-inflammatory effects of statins may result from reduced Rac1 prenylation, which may also explain some of our findings.

Since Henriksbo et al. reported an NLRP3 inflammasome-dependent production of IL1β by fluvastatin in BMMs, we aimed to reproduce their findings [46]. Our data do not support an NLRP3-dependent IL1β release upon treatment of either simvastatin or cerivastatin, which, again, might be due to differences regarding the treatment scheme.

Stress kinases, such as ERK, play an important role in macrophage-mediated inflammation. We, therefore, investigated ERK activation after statin treatment and found that statins induce ERK phosphorylation. ERK activation was previously observed by Lee et al. after simvastatin treatment in RAW 264.7 cells in a time-dependent manner [61]. Thus, the activation of ERK might in part contribute to statin-induced inflammatory effects.

Phagocytosis is an essential aspect of macrophage host defense and plays a crucial role in all stages of atherogenesis, either by LDL clearance and foam cell formation or regarding plaque stability. Moreover, phagocytosis is pivotal for pathogen clearance in a wide range of infectious diseases [40]. We found that statins decreased the phagocytotic activity in otherwise unstimulated cells and under short-term inflammatory conditions. These findings are in accordance with published data for human macrophages [20], murine peritoneal macrophages, and human monocytes [62]. Interestingly, other experimental setups with in vivo and ex vivo murine peritoneal macrophages suggested an enhanced phagocytotic activity upon statin treatment, which might be related to differences regarding the macrophage origin or the phagocytosed material [63,64]. Mechanistically, statins may impair phagocytosis via reduced isoprenylation of members of the Rho family of small G-proteins, as shown in previous studies [65,66].

The recently registered drug bempedoic acid was known as ESP-55016 and ETC-1002 when it was first synthesized in 2004 [67]. Due to the prodrug properties of bempedoic acid, i.e., the fact that its pharmacologically active form is generated by a liver-specific enzyme, its action is supposed to be limited to hepatic cells. In fact, clinical trials show no muscle-related effects, which are associated with statin therapy as adverse effects [68]. Still, despite its supposed liver-specific action, bempedoic acid has been described to exhibit anti-inflammatory activities [69]. A previous study showed reduced TNF levels in human LPS-treated macrophages after treatment with 50–100 µM bempedoic acid. The authors suggested a modulation of AMPK and MAPK pathways as the underlying mechanism [33]. The study also showed that macrophages did not convert the compound into its active form, implying that target-independent unspecific actions of the prodrug caused the observed effects.

While the *Slc27* gene family generally represents a family of fatty acid transporters [70], the protein encoded by *Slc27a2* also exhibits metabolic activity. This isozyme represents a member of the long-chain fatty-acid-coenzyme A ligase family, shows a preference for generating CoA derivatives of n-3 fatty acids [71], and is required to transform bempedoic acid into its active form [34]. We showed that different types of human macrophages express very low levels of *SLC27A2*, thereby supporting previous findings [33]. Furthermore, our data suggest that murine macrophages are also unable to transform the prodrug since *Slc27a2* was virtually not expressed in these cells.

In contrast to the study by Filippov et al. [33] we did not observe any effect of bempedoic acid for most readout parameters. This observation might be related to the fact that we used a lower concentration of 25 µM due to the toxicity of the compound itself (at ≥50 µM in RAW 264.7 cells) or the vehicle DMSO at the corresponding concentrations (at ≥0.5% in BMMs). The maximal serum concentration of bempedoic acid in vivo averages out at 60–90 µM, but the compound shows a high degree of plasma protein binding (>99%), which limits its activity [72,73]. Thus, less than 1 µM free bempedoic acid is available to affect immune cells in vivo, suggesting that a concentration of 25 µM should be sufficient to uncover potential effects in vitro.

Interestingly, the only clear effect of bempedoic acid was the rescue of the phagocytotic activity during its M2-associated decline. Since bempedoic acid is not converted into the ATP-citrate lyase-inhibitor bempedoic acid-CoA in macrophages [31], and macrophage cholesterol levels are not affected by bempedoic acid (Appendix A), a cholesterol-dependent mechanism can be ruled out. One might speculate that this effect is due to the enhanced fatty acid oxidation in bempedoic acid-treated cells, which plays a more prominent role during M2 polarization compared to M1 [67,74,75]. As mentioned above, it has been shown that bempedoic acid in its prodrug form can activate AMPK [31], and AMPK activation may enhance phagocytosis [76]. Thus, AMPK activation may represent another pathway by which bempedoic acid modulates the phagocytotic capacity of M2 macrophages.

Modulation of macrophage phenotypes has been suggested as a novel strategy for the pharmacological treatment of atherosclerosis. Macrophages with different functional phenotypes are likely to play different roles in the pathogenesis and progression of atherosclerosis: M1 macrophages have been associated with initiating and sustaining inflammation, and M2 macrophages have been linked to inflammation resolution. In fact, M2 macrophages are particularly abundant in stable zones of the plaque and asymptomatic lesions. However, a broad spectrum of intermediate phenotypes has been identified in vivo studies. Different stimuli, such as various cytokines, lipids, senescent or apoptotic cells, and iron, can influence macrophage phenotypes in atherosclerotic lesions, thus generating a complex microenvironment that cannot be fully recapitulated in in vitro studies [77,78].

Our data suggest detrimental pro-inflammatory effects of statins on macrophages within the plaque, although another inflammatory trigger may be required. Indeed, it has been reported that full plaque regression upon treatment with cholesterol-lowering agents may be prevented if macrophage inflammation persists [78]. Our data show that an inflammatory activation might be fueled by statins, thus potentially limiting plaque regression. However, this issue might be outweighed by the beneficial effect of statins on other cell types, such as endothelial and smooth muscle cells, and the overall impact of reduced serum cholesterol levels [79].

In other contexts, statins may affect macrophages in a favorable manner due to their ability to induce KLF2: a recent study revealed that myeloid KLF2 reduces metabolic inflammation in peripheral and central tissues in a mouse model of obesity [32]. Thus, the impact of statin treatment on macrophages in vivo most likely depends on the degree of inflammation within the microenvironment.

Statins have been suggested as potential therapeutics for diseases beyond CVD, particularly inflammatory lung diseases and infectious diseases [11,12]. In our hands, statin treatment led to hyperinflammation under conditions that mimic acute inflammation, i.e., short-term LPS treatment. The decreased phagocytotic capacity in statin-treated cells implies that the clearance of pathogens in patients undergoing statin therapy may be reduced. Of note, phagocytosis can also contribute to disease progression if internalized bacteria are not completely killed, and the bactericidal capacity of macrophages is linked to a pro-inflammatory phenotype [80]. Thus, the pro-inflammatory state induced by statins may contribute to a better outcome in bacterial infections, which was previously observed for patients on statin therapy [13,14]. Bempedoic acid, on the other hand, had negligible effects on inflammation, but even enhanced phagocytosis, at least in M2 macrophages. Again, the outcome of bempedoic acid treatment during a bacterial infection would depend on whether their bactericidal capacity is also altered. Thus, it would be interesting to examine the influence of bempedoic acid and statins on the bactericidal activity of macrophages in future studies.

Taken together, our data point towards a unique cholesterol-independent modulation of macrophage functions by statins, which is not exhibited by bempedoic acid. Furthermore, our data suggest that the anti-inflammatory properties that statins and bempedoic acid show in vivo are not related to their effects on macrophages but are more likely linked to systemic effects or effects on other cell types.

## 4. Materials and Methods

### 4.1. Reagents

Cell media (RPMI1640, #R0883; DMEM, #D6546), fetal calf serum (FCS, #F7524), penicillin/streptomycin (#P433), and glutamine (#G7513) were purchased from Sigma-Aldrich (St. Louis, MO, USA). PAN-FCS (#P040-37500) was purchased from PAN-Biotech (Aidenbach, Germany). Zeocin^TM^ (#ant-zn-05), Normocin^TM^ (#ant-nr-1) and HEK-Blue^TM^ Selection (#hb-sel) were purchased from InvivoGen (San Diego, CA, USA). Anti-p44/42 (ERK1/2) mouse antibody (L34F12, #4696S) and anti-phospho-p44/42 MAPK (Thr202/Tyr204) rabbit mAbs (20G11, #4376S) were obtained from Cell Signaling Technology (Danvers, MA, USA). Anti-rabbit IRDye 680- and anti-mouse IRDye 800-conjugated secondary antibodies were from LI-COR Biosciences (#926-68071, #926-32210) (Lincoln, NE, USA). Ultrapure LPS from *E. coli* K12 (#tlrl-peklp) and QUANTI-Blue™ (#rep-qb) were purchased from InvivoGen (San Diego, CA, USA). MTT (# M5655), actinomycin D (#A9415), PD98059 (#P215), cerivastatin sodium salt hydrate (#SML0005), and (R)-Mevalonic acid lithium salt (#50838) were obtained from Sigma-Aldrich (St. Louis, MO, USA). Murine M-CSF (#130-101-704), IFNγ (#130-105-782), IL4 (#130-094-061), and Il1β (#130-101-681) were obtained from Miltenyi Biotech (Bergisch Gladbach, Germany). Simvastatin sodium salt (#10010345) was purchased from Cayman Chemicals (Ann Arbor, MA, USA) and Bempedoic acid (#738606-46-7) was purchased from MedChemExpress (Monmouth Junction, NJ, USA). 5xHotFirePOl EvaGreen qPCR Mix (no Rox) (#08-25-00001) was purchased from Soli Biodyne (Tartu, Estland). Rockland Blocking Buffer (#MB-070) was purchased from Biomol (Hamburg, Germany). pHrodo™ Red *S. aureus* Bioparticles™ Conjugate for Phagocytosis (#A10010) was purchased from Thermo Fisher Scientific (Waltham, MA, USA). Other chemicals were obtained from either Sigma-Aldrich (St. Louis, MO, USA) or Carl Roth (Karlsruhe, Germany) unless stated otherwise.

### 4.2. Cell Culture

RAW-Blue^TM^ cells (InvivoGen, San Diego, CA, USA, #raw-sp) were grown in high glucose DMEM medium supplemented with 10% heat-inactivated FCS (30 min at 56 °C), 2 mM glutamine, 100 U/mL penicillin G, 100 µg/mL streptomycin, 100 µg/mL Normocin, and 200 µg/mL Zeocin for selection.

HEK-Blue^TM^ IL-1R cells (InvivoGen, San Diego, CA, USA, #hekb-il1r) were grown in high-glucose DMEM medium supplemented with 10% heat-inactivated FCS (30 min at 56 °C), 2 mM glutamine, 100 U/mL penicillin G, 100 µg/mL streptomycin, 100 µg/mL Normocin, and 1 × HEK-Blue^TM^ Selection.

L929 cells and RAW 264.7 cells (American Type Culture Collection) were cultivated in standard medium (RPMI 1640, 10% FCS, 100 U/mL penicillin G, 100 µg/mL streptomycin, 2 mM glutamine). The cells were maintained at 37 °C in a humidified atmosphere of 5% CO_2_.

BMMs were obtained from wild-type (WT) or *Nlrp3* knockout (KO) mice as described previously [47]. Femurs and tibias were flushed with standard medium (RPMI 1640, 10% PAN-FCS, 100 U/mL penicillin G, 100 µg/mL streptomycin, 2 mM glutamine). After centrifugation (10 min, 200× *g*), erythrocytes were lysed by incubation in hypotonic buffer (155 mM NH_4_Cl, 10 mM KHCO_3_, 1 mM Na_2_EDTA) for 3 min at 37 °C. Cells were washed with PBS, resuspended in standard medium containing M-CSF (50 ng/mL, 30 mL per preparation), transferred into a 75 cm^2^ culture flask, and cultured overnight. Fibroblast-like cells, mature mononuclear phagocytes, and other cells adhering to the flask were discarded [81]. Non-adherent cells were collected and cultured in a 150 cm^2^ culture flask for another 5 to 6 d in M-CSF-containing medium. Differentiated cells were detached with Accutase^®^ (Sigma-Aldrich, St. Louis, MO, USA #A6964), suspended in standard medium supplemented with 50 ng/mL M-CSF, and seeded into 96-well plates (7.5 × 10^4^ cells/well) for TNF and IL1β measurements as well as MTT assays and 5.0 × 10^4^/well for phagocytosis and Griess assays or into 24-well plates (2.5 × 10^5^ cells/well) for RT-qPCR and Amplex^®^ cholesterol kit (Thermo Fisher Scientifc, Waltham, MA, USA, #A12216) and into 6-well plates (10^6^/well) for Western blot analysis. The solvent control was 0.25% DMSO. Higher DMSO concentrations were avoided due to toxicity issues (at 0.5%: viability reduced by 18.7 ± 3.4%, *p* = 0.0004, determined by MTT assay).

### 4.3. Mice

Mice were housed in a 12:12 h light–dark cycle with food and water ad libitum. For all experiments, C57B/6 mice of the same age were used. For the IL1β bioassay, we used mice in which the entire coding sequence of *Nlrp3* was replaced with a Neo cassette (*Nlrp3* KO mice, The Jackson Laboratory, Bar Harbor, ME USA; #B6.129S6- Nlrp3^tm1Bhk^/J). Genotyping was performed with 5x HOT FIREPol EvaGreen^®^ qPCR Mix and a total volume of 20 µL and primer sequences as follows: mutant forward 5′-TGCCTGCTCTTTACTGAAGG-3′, wild type forward 5′-TCAGTTTCCTTGGCTACCAGA-3′, and common reverse 5′-TTCCATTACAGTCACTCCAGATGT-3′, as described by The Jackson Laboratory. B6.129P2-Lyz2^tm1(cre)Ifo^/J mice (The Jackson Laboratory) were crossed with C57B/6 mice bearing LoxP sites upstream and downstream of *Tsc22d3* exon 6 to obtain myeloid-specific *Gilz* knockout (KO) mice. Breeding and genotyping were performed as described previously [47].

### 4.4. Endotoxin Quantification

Simvastatin, cerivastatin, and bempedoic acid were tested for the absence of endotoxin using the endotoxin assay PyroGene^TM^ Assay (Lonza, Basel, Switzerland, #50-658U) according to the manufacturer’s instructions. Its sensitivity ranges from 0.0005 to 5 EU/mL. Briefly, 100 µL aliquots of samples diluted in endotoxin-free water and tested in the highest used concentration as well as standards were transferred to a microtiter plate. Each sample was measured in duplicate, and a standard curve was run alongside the samples. After 60 min incubation at 37 °C, fluorescence was measured using a fluorescence reader (Promega^TM^ GloMax^®^ Plate Reader, Madison, WI, USA) at 415–445 nm emission and 365 nm excitation. Endotoxin levels of final compound concentrations were below those of cell culture media and sera.

### 4.5. Cytotoxcity Measurements

To ensure the use of non-toxic concentrations of all compounds, simvastatin, cerivastatin, bempedoic acid, PD98059, and mevalonate, the MTT colorimetric assay was performed as described previously (Appendix A) [47]. Absorbance was measured at 560 nm using a microplate reader (Promega^TM^ GloMax^®^ Plate Reader Madison, WI, USA). The cell viability obtained from at least three independent experiments performed in triplicate or sextuplicate was calculated relative to solvent controls.

### 4.6. Gene Expression in Human Macrophages

Publicly available RNA sequencing data sets (Gene Expression Omnibus (GEO) datasets GSE162669 and GSE162698) were analyzed to assess *HMGCR*, *ACLY*, and *SLC27A2* expression in human monocyte-derived macrophages and human alveolar macrophages, as detailed in [44].

### 4.7. RNA Isolation, Reverse Transcription, and Quantitative PCR (RT–qPCR)

Quantitative RT–PCR (qPCR) was performed as described previously [47]. Total RNA from cells was isolated using the High Pure RNA Isolation Kit (Roche, Basel, Switzerland, #11828665001), following the manufacturer’s instructions. RNA was reverse transcribed using the High-Capacity cDNA Reverse Transcription Kit (Thermo Fisher Scientific, #4368813) in the presence of the RNase inhibitor RNaseOUT™ (Thermo Fisher Scientific, #10777019) following the manufacturer’s instructions. qPCR was performed using the 5x HOT FIREPol EvaGreen^®^ qPCR Mix and a total volume of 20 µL. The primer sequences for each transcript are detailed in Table 1. For each primer pair, an annealing temperature of 60 °C was used (except *Nlrp3* with 59 °C annealing temperature). The CFX96 touch™ Real-Time PCR detection system (Bio-Rad Laboratories, Hercules, CA, USA) was used to quantify gene expression. Data were analyzed with the comparative ΔΔCt method. The housekeeping gene was chosen after evaluating the expression stability of at least three candidate genes under the experimental conditions, using the geNorm, NormFinder, and BestKeeper Software tools [82]. Absolute amounts of the transcript were normalized to the corresponding housekeeping genes.

### 4.8. Western Blot

Cells were lysed in lysis buffer (50 mM Tris-HCl, 1% (*m*/*v*) SDS, 10% (*v*/*v*) glycerol, 5% (*v*/*v*) 2-mercaptoethanol, 0.004% (*m*/*v*) bromphenol blue) supplemented with a protease inhibitor mix (cOmplete^®^; Roche Diagnostics, Basel, Switzerland, #04693124001) and stored at −80 °C until further use. After sonication, lysates were boiled for 5 min at 95 °C. Proteins were separated by SDS-PAGE on 12% gels using the Mini-Protean Tetra Cell system (BioRad, Hercules, CA, USA) and transferred onto Immobilon FL-PVDF membranes (Millipore, Burlington, MA, USA, #IPFL00010) using the Tetra Blotting Module (BioRad). The membranes were blocked in Rockland blocking buffer for near-infrared Western Blotting for 1.5 h, incubated with primary antibody dilutions (1:1000 in Rockland blocking buffer) at 4 °C overnight and with IRDye 680 or IRDye 800 conjugated secondary antibodies (1:10,000 in Rockland blocking buffer) for 1.5 h at room temperature. Blots were scanned with an Odyssey Infrared Imaging System (LI-COR Bioscience, Lincoln, NE, USA), and relative protein amounts were determined using the Odyssey software.

### 4.9. Cytokine Quantification

Murine TNF was quantified by bioassay as previously described [83]. L929 cells were seeded at a density of 3 × 10^4^ cells/well into a 96-well plate. After 24 h, the medium was replaced by 100 µL of actinomycin D solution (1 µg/mL in standard medium), and cells were incubated for 1 h at 37 °C. Subsequently, BMM supernatants (1:5 diluted, 100 µL total volume/well) were added. Dilution series of recombinant murine TNF (100–2500 pg/mL) were run alongside the samples to generate a standard curve. The plates were incubated for an additional 24 h at 37 °C. The MTT assay was used to assess cell viability.

Murine IL1β was quantified by bioassay. HEK-Blue^TM^ IL-1R cells were seeded at a density of 5 × 10^4^ cells/well with BMM supernatants (1:100 diluted, 20 µL total volume/well) into a 96-well plate and incubated overnight at 37 °C. Dilution series of recombinant murine IL1β (10^−5^–10^1^ ng/mL) were run alongside the samples to generate a standard curve. The next day, supernatants were collected, and secreted embryonic alkaline phosphatase (SEAP) activity was determined using the QUANTI-Blue^TM^ Solution according to the supplier’s instructions. SEAP levels were determined at 600 nm with a microplate reader (Promega^TM^ GloMax^®^).

### 4.10. NF-κB/AP-1 Reporter Cells

RAW-Blue^TM^ cells (InvivoGen, San Diego, CA, USA, #raw-sp) expressing SEAP under the control of the IFNβ minimal promoter fused to five NF-κB and AP-1 binding sites were used to determine NF-κB/AP-1 activity. Cells were seeded into 96-well plates (10^5^ cells/well) and treated as indicated. After 24 h, supernatants were collected, and SEAP activity was determined using the QUANTI-Blue^TM^ Solution according to the supplier’s instructions. SEAP levels were determined at 600 nm with a microplate reader (Promega^TM^ GloMax^®^).

### 4.11. Griess Assay

BMMs were cultured in 96-well plates (5 × 10^5^ cells/well) and treated as indicated. After 24 h, the concentration of nitrite was measured in the supernatants by Griess assay as previously described [84]. In brief, 90 µL 1% sulfanilamide in 5% H_3_PO_4_ and 90 µL 0.1% N-(1-naphthyl)ethylenediamine dihydrochloride in H_2_O were added to 100 µL of cell culture supernatant, followed by absorbance measurement at 560 nm using a Promega^TM^ GloMax^®^ Plate Reader. A standard curve of sodium nitrite dissolved in medium was run alongside the samples.

### 4.12. Cholesterol Quantification

Intracellular cholesterol (free cholesterol and cholesteryl esters) was quantified with the Amplex^®^ Red Cholesterol Assay Kit (Thermo Fisher Scientifc, #A12216) according to the manufacturer’s instructions with minor modifications as detailed in [44]. Briefly, BMMs were lysed in 200 µL reaction buffer. A total of 50 µL of a 1:5 dilution of the lysate in sample puffer were transferred into a 96-well plate and incubated with the Amplex^®^ working solution. Each sample was measured in triplicate, and a standard curve was run alongside the samples. After 30 min incubation (37 °C, protected from light), fluorescence was measured using a plate reader (Promega^TM^ GloMax^®^) at 580–640 nm emission and 520 nm excitation. Total cellular protein concentrations used for data normalization were determined by Pierce BCA protein assay (Thermo Fisher Scientific, #23225) according to the manufacturer’s instructions.

### 4.13. Phagocytotic Activity

BMMs were seeded into a 96-well plate (5 × 10^4^ cells/well, 100 µL) and treated as indicated. After 24 h treatment, 5 µg pHrodo™ Red *S. aureus* Bioparticles™ Conjugate were added. Cells were imaged for 24 h in an IncuCyte S3 system. The phagocytotic activity was analyzed with the IncuCyte analysis software and expressed as mean red fluorescence intensity normalized to confluence.

### 4.14. Statistical Analysis

All experiments were performed at least three times. Data distribution was determined by the Shapiro–Wilk test. One-sample *t*-test followed by a Bonholm post hoc test was used for analyzing gene expression data of the control group and Western blot. Means of more than two groups were compared by one-way ANOVA with a Bonholm post hoc test (normal distribution). Means of more than two groups that have been split into two independent variables were compared by two-way ANOVA with a Bonholm post hoc test. Statistical significance was set as * *p* < 0.05, ** *p* < 0.01, and *** *p* < 0.001 compared with controls or as indicated. Data analysis was performed using Origin software (OriginPro 2018b; OriginLab, Northampton, MA, USA).

## Figures and Tables

**Figure 1 ijms-22-12480-f001:**
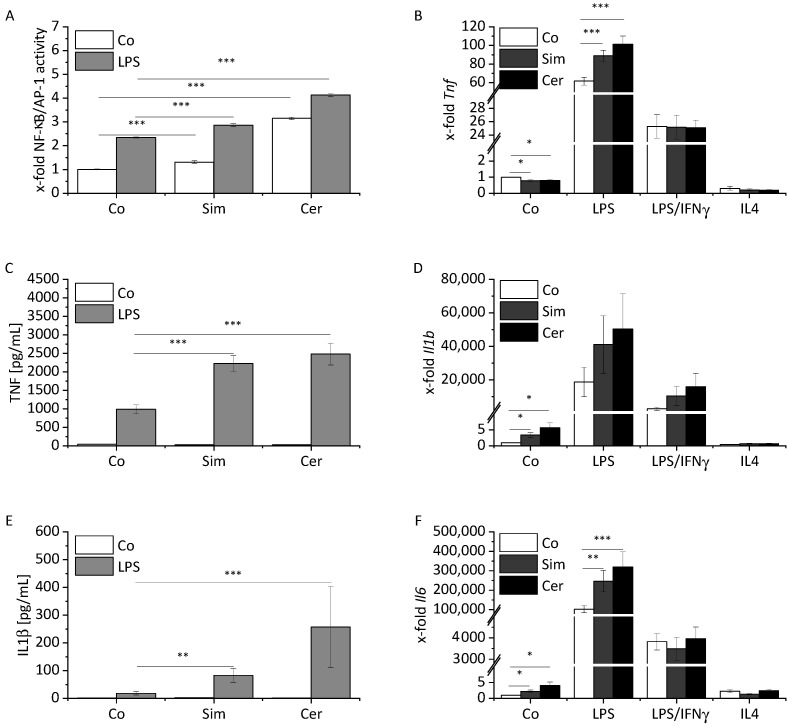
Effect of statin treatment on inflammatory macrophage activation. (**A**) RAW-Blue^TM^ cells were treated with either simvastatin (Sim, 2 µM) or cerivastatin (Cer, 1 µM) for 24 h. Inflammatory activation was induced by treatment with LPS (100 ng/mL) for the final 4 h. NF-κB/AP-1 activity was determined by secreted embryonic alkaline phosphatase (SEAP) detection. Co = solvent control (*n* = 3, triplicates). (**B**,**D**,**F**,**H**) *Tnf* (**B**), *Il1b* (**D**), *Il6* (**F**), and *Nos2* (**H**) mRNA expression in BMMs was determined by real-time RT-PCR, normalized to *Ppia*, and expressed as x-fold of Co. BMMs were stimulated for the last 4 h with LPS (100 ng/mL) or polarized towards M1 (LPS, 100 ng/mL; IFNγ, 20 ng/mL), or M2 (IL4, 20 ng/mL) in the presence or absence of Sim (2 µM) or Cer (0.5 µM) for 24 h. Co = solvent control (*n* = 6). (**C**,**E**) TNF (**C**) and IL1β (**E**) were measured by bioassay. BMMs were treated for 24 h with either Sim (2 µM) or Cer (0.5 µM). Inflammatory activation was induced by treatment with LPS (100 ng/mL for IL1β, 10 ng/mL for TNF) for the final 4 h. Co = solvent control (*n* = 3, duplicates for TNF, triplicates for IL1β). (**G**) Nitrite production was measured by Griess assay. BMMs were treated for 24 h with either Sim (2 µM) or Cer (0.5 µM). Samples were stimulated for the final 20 h (LPS, 50 ng/mL; IFNγ, 20 ng/mL). Co = solvent control (*n* = 3, triplicates). A one-sample *t*-test followed by a Bonholm post hoc test was used for analyzing gene expression data of the control group. Means of more than two groups were compared by one-way ANOVA with Bonholm post hoc test (normal distribution). * *p* < 0.05, ** *p* < 0.01, and *** *p* < 0.001.

**Figure 2 ijms-22-12480-f002:**
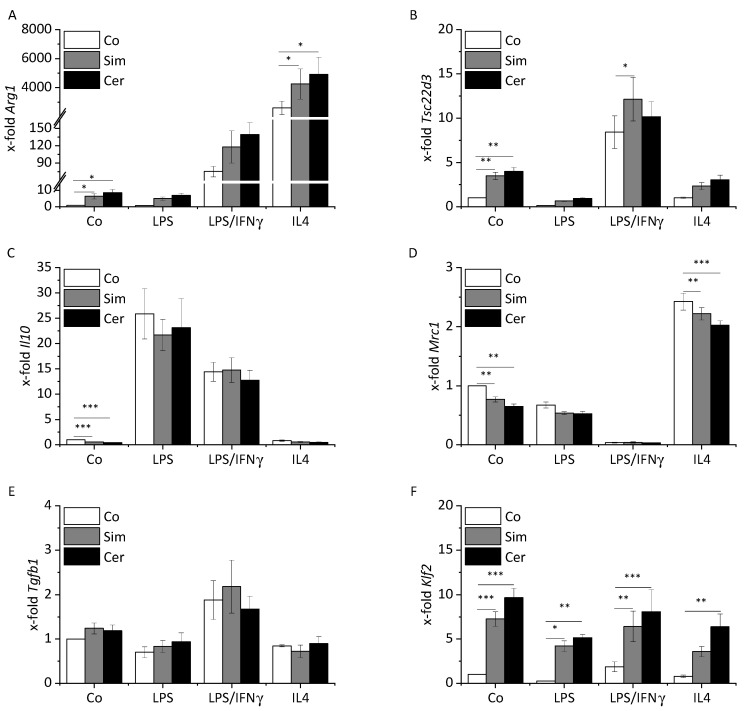
Effect of statin treatment on the anti-inflammatory response in macrophages. (**A**–**D**) *Arg1* (**A**), *Tsc22d3* (**B**), *Il10* (**C**), *Mrc1* (**D**), *Tgfb1* (**E**), and *Klf2* (**F**) mRNA expression in BMMs was determined by real-time RT-PCR, normalized to *Ppia*, and expressed as x-fold of Co. BMMs were stimulated for the last 4 h with LPS (100 ng/mL) or polarized towards M1 (LPS, 100 ng/mL; IFNγ, 20 ng/mL) or M2 (IL4, 20 ng/mL) in the presence or absence of of simvastatin (Sim, 2 µM) or cerivastatin (Cer, 0.5 µM) for 24 h. Co = solvent control (*n* = 6). A one-sample *t*-test followed by a Bonholm post hoc test was used for analyzing the gene expression data of the control group. Means of more than two groups were compared by one-way ANOVA with a Bonholm post hoc test (normal distribution). * *p* < 0.05, ** *p* < 0.01, and *** *p* < 0.001.

**Figure 3 ijms-22-12480-f003:**
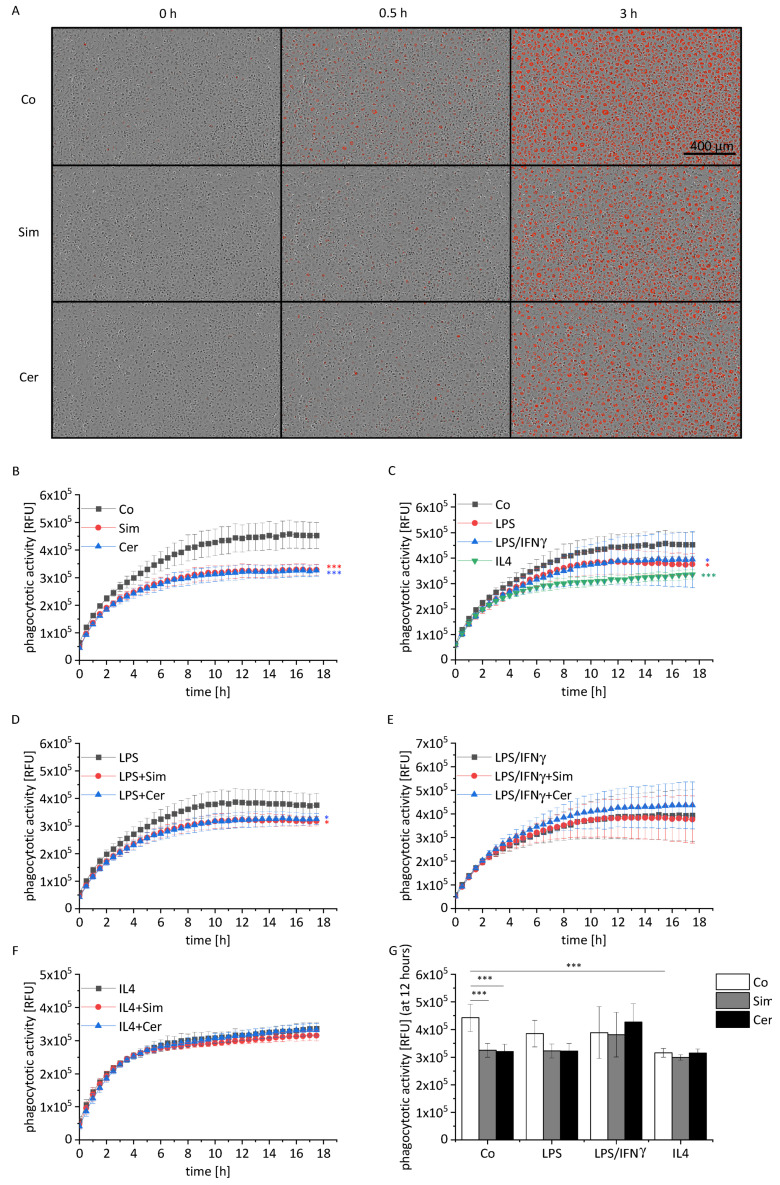
Effect of statin treatment on the phagocytotic activity of macrophages. (**A**–**G**) BMMs were stimulated for the last 4 h with LPS (100 ng/mL) or polarized towards M1 (LPS, 100 ng/mL; IFNγ, 20 ng/mL), or M2 (IL4, 20 ng/mL) in the presence or absence of simvastatin (Sim, 2 µM) or cerivastatin (Cer, 0.5 µM) for 24 h and monitored by an IncuCyte S3 system after addition of fluorogenic pHrodo^®^ Red *S. aureus* bioparticles (5 µg/well). Co = solvent control, RFU = relative fluorescence units. (**A**) Representative pictures at indicated time points are shown. (**B**–**G**) Quantification of phagocytotic activity expressed as mean red fluorescence intensity normalized to confluence (*n* = 4, duplicates). Statistical analysis was performed by two-way ANOVA with Bonholm post hoc test (**B**–**F**) or one-way ANOVA with Bonholm post hoc test (**G**). * *p* < 0.05, *** *p* < 0.001.

**Figure 4 ijms-22-12480-f004:**
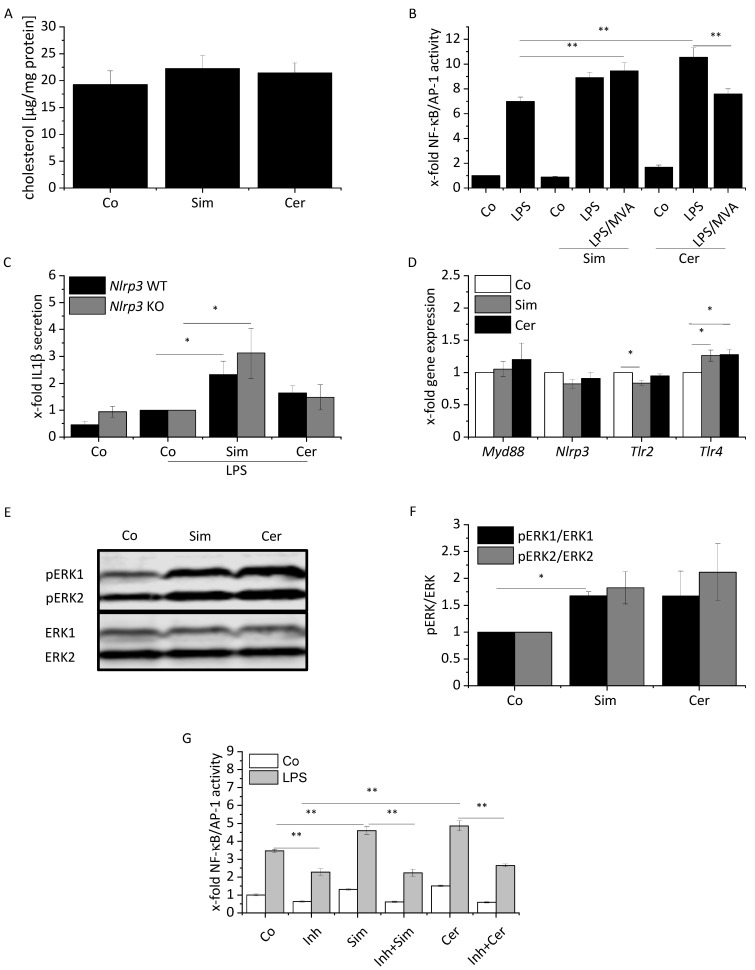
Statins affect different signaling pathways in macrophages. (**A**) Intracellular cholesterol levels. BMMs were either treated for 24 h with simvastatin (Sim, 2 µM) or cerivastatin (Cer, 0.5 µM). Co = solvent control (*n* = 3, triplicates). (**B**) RAW-Blue^TM^ cells were treated for 24 h with either Sim (2 µM) or Cer (1 µM). Cells were co-treated with mevalonate (MVA, 100 µM) where indicated. Inflammatory activation was induced by treatment with LPS (100 ng/mL) for the final 4 h. NF-κB/AP-1 activity was determined by secreted embryonic alkaline phosphatase (SEAP) detection. Co = solvent control (*n* = 3, triplicates). (**C**) IL1β was measured by bioassay. BMMs of *Nlrp3* WT and KO BMMs were treated for 24 h with either Sim (2 µM) or Cer (0.5 µM). Inflammatory activation was induced by treatment with LPS (100 ng/mL) for the final 4 h. Co = solvent control (*n* = 4 each WT and KO, quadruplicates). (**D**) mRNA expression of indicated genes were determined by real-time RT-PCR, normalized to *Ppia*, and expressed as x-fold of Co. BMMs were either treated for 24 h with Sim (2 µM) or Cer (0.5 µM). Co = solvent control (*n* = 6). (**E**,**F**) ERK phosphorylation was examined by Western Blot analysis. BMMs were treated with Sim (2 µM) or Cer (0.5 µM) for one hour. Co = solvent control. (**E**) One representative blot is shown. (**F**) Signal intensities were quantified and normalized to total ERK (*n* = 3). (**G**) RAW-Blue^TM^ cells were pre-treated for 30 min with the ERK inhibitor PD98059 (Inh, 10 µM). Cells were treated for 24 h with either Sim (2 µM) or Cer (1 µM). Inflammatory activation was induced by treatment with LPS (100 ng/mL) for the final 4 h. NF-κB/AP-1 activity was determined by secreted embryonic alkaline phosphatase (SEAP) detection. Co = solvent control (*n* = 3, triplicates). One-sample *t*-test followed by Bonholm post hoc test was used for analyzing gene expression data of the control group and Western blot (**D**,**F**). Means of more than two groups were compared by one-way ANOVA with Bonholm post hoc test (normal distribution) (**B**,**C**,**G**). * *p* < 0.05, ** *p* < 0.01.

**Figure 5 ijms-22-12480-f005:**
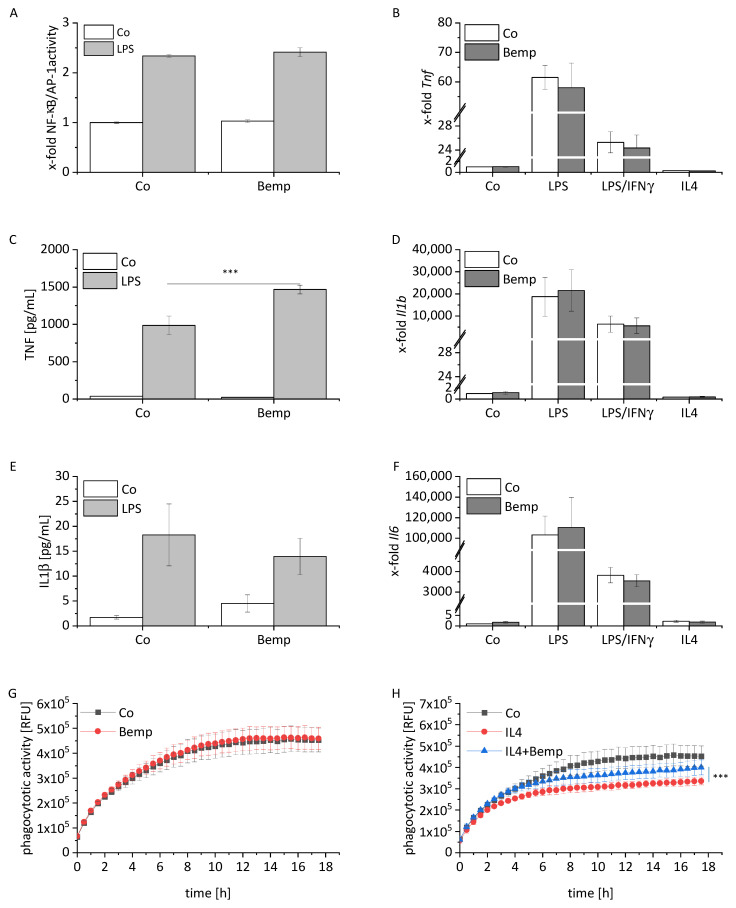
Effect of bempedoic acid treatment macrophages. (**A**) RAW-Blue^TM^ cells were treated with bempedoic acid (Bemp, 25 µM) for 24 h. Inflammatory activation was induced by treatment with LPS (100 ng/mL) for the final 4 h. NF-κB/AP-1 activity was determined by secreted embryonic alkaline phosphatase (SEAP) detection. Co = solvent control (*n* = 3, triplicates). (**B**,**D**,**F**) *Tnf* (**B**), *Il1b* (**D**), and *Il6* (**F**), mRNA expression in BMMs was determined by real-time RT-PCR, normalized to *Ppia*, and expressed as x-fold of Co. BMMs were stimulated for the last 4 h with LPS (100 ng/mL) or polarized towards M1 (LPS, 100 ng/mL; IFNγ, 20 ng/mL) or M2 (IL4, 20 ng/mL) in the presence or absence of Bemp (25 µM) for 24 h. Co = solvent control (*n* = 6). (**C**,**E**) TNF (**C**) and IL1β (**E**) were measured by bioassay. BMMs were treated for 24 h with Bemp (25 µM). Inflammatory activation was induced by treatment with LPS (100 ng/mL for IL1β, 10 ng/mL for TNF) for the final 4 h. Co = solvent control (*n* = 3, duplicates for TNF, triplicates for IL1β). (**G**,**H**) BMMs were treated for 24 h with Bemp (25 µM) in the presence or absence of IL4 (20 ng/mL) and monitored by an IncuCyte S3 system after the addition of fluorogenic pHrodo^®^ Red *S. aureus* bioparticles (5 µg/well). Quantification of phagocytotic activity expressed as mean red fluorescence intensity normalized to confluence (*n* = 4, duplicates). RFU = relative fluorescence units. A one-sample t-test followed by a Bonholm post hoc test was used for analyzing gene expression data of the control group. Means of more than two groups were compared by one-way ANOVA with Bonholm post hoc test (normal distribution). Statistical analysis of phagocytotic activity was performed by two-way ANOVA with Bonholm post hoc test. *** *p* < 0.001.

**Table 1 ijms-22-12480-t001:** Primer sequences for RT–qPCR analyses.

Gene	Accession Number	Forward Primer Sequence 5′-3′	Reverse Primer Sequence 5′-3′
*Acly*	NM_134037.3	ATGCCCCAAGGAAAGAGTGC	CTCGGGAACACACGTAGTCA
*Arg1*	NM_007482.3	ACAAGACAGGGCTCCTTTCAG	GGCTTATGGTTACCCTCCCG
*Hmgcr*	NM_008255.2	ATCCAGGAGCGAACCAAGAGAG	CAGAAGCCCCAAGCACAAAC
*Il10*	NM_010548.2	GCCCAGAAATCAAGGAGCAT	GAAATCGATGACAGCGCCT
*Il6*	NM_031168.2	AAGAAATGATGGATGCTACCAAACTG	GTACTCCAGAAGACCAGAGGAAATT
*Klf2*	NM_008452.2	CCTTGCACATGAAGCGACAC	ACTTGTCCGGCTCTGTCCTA
*Myd88*	NM_010851.3	TAAGTTGTGTGTGTCCGACCG	CATGCGGCGACACCTTTTCT
*Nos2*	NM_010927.3	CTTCCTGGACATTACGACCC	TACTCTGAGGGCTGACACAA
*Ppia*	NM_008907.1	GCGTCTCCTTCGAGCTGTTT	CACCCTGGCACATGAATCCT
*Slc27a2*	NM_011978.2	AGCGGAGAGACCTCCTGATGAT	CAGAAGCCCCAACAAGCACAAAC
*Tlr2*	NM_011905.3	CACTGCCCGTAGATGAAGTC	TACCTCCGACAGTTCCAAGA
*Tlr4*	NM_021297.3	TCCCTGCATAGAGGTAGTTCC	TCAAGGGGTTGAAGCTCAGA
*Tnf*	NM_013693.2	CCATTCCTGAGTTCTGCAAAGG	AGGTAGGAAGGCCTGAGATCTTATC
*Tgfb1*	NM_011577.1	ACCCTGCCCCTATATTTGGA	CGGGTTGTGTTGGTTGTAGAG
*Tsc22d3*	NM_010286.4	GCTGCTTGAGAAGAACTCCCA	GAACTTTTCCAGTTGCTCGGG

## Data Availability

Publicly available RNA sequencing data sets are accessible via the Gene Expression Omnibus (GEO) database (GSE162669 and GSE162698). Other datasets utilized in the present study are available from the corresponding author on reasonable request.

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
