# Peer review of "Statins and Bempedoic Acid: Different Actions of Cholesterol Inhibitors on Macrophage Activation"

_ijms, 2021, doi:10.3390/ijms222212480_

Round 1

Reviewer 1 Report

Rebecca Linnenberger et al have provided a revised manuscript where they clarified some aspects. The manuscript still lacks solid conclusions and the ones reported by the authors are not supported by provided data.

  1. Authors never speculate on the proinflammatory effect of statins they see in short term LPS. They should comment on how this may affect (at all) a situation where statins are given when a chronic inflammation, insulin resistance and atherosclerosis are ongoing. Authors should consider using different models of activation, that could be more suited for the purpose on understanding what statins may do to macrophages. For example macrophages treated with oxLDL or other types of nonclassical M1 and M2 activation.
  2. Authors should use primary cells for their studies besides BMMs, like resident macrophages and thioglycolate-elicited macrophages, or macrophages isolated from mouse models of High Fat diet or atherosclerosis.
  3. The part of the manuscript about discrepancies in the literature is extremely confusing. Authors should get to a consensus of a validated model for studying the effect of statins in macrophages.
  4. Authors never make clear what phenotype of macrophages may be important to achieve (or maintain) during hypercholesterolemia or cardiovascular diseases for amelioration of disease. What phenotype would be desired to delay progression/ameliorate pathologic condition?
  5. Authors should describe better “metabolic inflammation”.
  6. It is still not clear the mechanism by which statins act on calcium homeostasis in the muscle.
  7. Conclusion of paragraph 2.1 is hard to understand since they talk about a “unique statin-induced modulation of the macrophage phenotype” when a significant effect is seen only in one of the 3 types of polarizations tested.Authors should be more specific.
  8. It is counterintuitive that authors register a decrease in phagocytic activity during LPS stimulation, when it is conceivable that it should increase during pro-inflammatory activation. Authors should provide explanation for this and cite literature.
  9. Statistical symbols of Fig3G need to be better explained in the legend, especially about comparisons between groups.
  10. Authors still do not provide valid explanations for MVA -driven effect.
  11. Authors should provide reference about BMMs synthesizing cholesterol in unstimulated condition and in regular cell culture media as this is highly unlikely. Moreover, authors should measure it with the polarization stimuli + and – statins.
  12. It would be important that authors worked on reproducibility of their experiments: it is not clear why in fig4C they see a significant increase of iL1b with LPS+Sim when they do not see the same in Fig1D. It is not clear if authors are getting different results from Raw cells vs BMMs.
  13. Authors should explain the effect of the ERK inhibitor in Fig 4G when in Cer+LPS-treated cells pERK/ERK ratio is not significantly different (fig 4E-F).
  14. Authors should consider using ERK inhibitor for the effect seen on phagocytosis as well.
  15. It would be important that authors explained why the effects mediated by statins they are reporting are cholesterol independent.
  16. Authors should speculate on the repercussions of bempedoic acid rescuing phagocytosis in IL-4 treated cells, from a disease-model point of view.

Reviewer 2 Report

This manuscript by Linnenberger et al. is interesting in that the authors investigate the role of statins on macrophage polarization. Since statins are the most prescribed drug, understanding their effects on macrophage function is critical. They found that macrophage phagocytosis is inhibited after treatment with simvastatin or cerivastatin and macrophages also displayed a pro-inflammatory phenotype after treatment with LPS. The inflammatory response was driven by an increase in ERK activation. Using bempedoic acid, which inhibits ATP-citrate lyase (a central metabolic enzyme that catalyzes the ATP-dependent conversion of citrate and coenzyme A (CoA) to oxaloacetate and acetyl-CoA), they show no alteration of macrophage polarization. The data is interesting and has implications in understanding the effect of statins on macrophage function. There are several concerns raised with the data presented, and these are summarized below.

  1. It is unclear if the cholesterol lowering drugs used in this study alter macrophage cholesterol levels, regulate lipidation of Rho/Rac family of small GTPases, or alter intermediates of the mevalonate pathway.
  2. The images in figure 3A are confusing. It appears that over the course of 3 h, the cells have greatly multiplied. Also, under all 3 conditions all the cells are staining red at 3 h. However, the quantification in 3B does not match what is shown in 3A.
  3. It is unclear to this reviewer if macrophages are not responding to bempedoic acid treatment or if bempedoic acid rescues macrophage phagocytic function. The authors need to clarify this.

Round 2

Reviewer 1 Report

Rebecca Linnenberger et al have provided data and nice clarifications to some of the reviewer’s comments. Although the authors have raised some interesting possibilities that may add positive value to the manuscript if further dissected, the conclusion is not strongly supported and does not reflect physio-pathological situations of macrophage behavior when statins are given.

  1. Although authors have pointed out that patients with chronic conditions may still encounter acute infections and inflammation, hence their model to study short LPS response, they do not further speculate on the data obtained, and what would be the result, from the inflammatory side, to keep individuals on statins.
  2. Authors have provided some nice data with oxLDL treatment of macrophages; however they measured only few markers during this treatment. The use of statins in the presence of oxLDL with LPS or other stimuli is absolutely valuable since statins are given in situation of metabolic syndrome and oxLDL treatment may be a good model for macrophage behavior in this situation.
  3. The lack of data on peritoneal macrophages is still a limitation of the study. Authors could use any mouse model with mice treated with statins and evaluate macrophage behavior/markers, isolating them from different sites.
  4. Mechanisms of statins altering calcium homeostasis in muscle cells is still not addressed.
  5. For suppl fig 10, authors should show raw values in order to appreciate the difference in the ctrl low serum vs the ctrl in 10%fbs.
  6. Although authors cite literature of M1/M2 macrophage polarization not affecting cellular cholesterol levels, they need to further explain the full mechanisms of interplay between biosynthesis vs uptake. Moreover, authors need to assess this aspect in a metabolic syndrome-type of situation. What happens to macrophages surrounded by lipid mixtures (oleate/palmitate etc)? Do they accumulate cholesterol or esters? Free fatty acids (FFA) are common feature in the plasma of individuals with metabolic syndrome so authors should investigate this situation (lipid mixture/lipid enriched media with LPS/other stimuli + statins). This would be important to really understand if the effect is cholesterol independent. Authors are assuming macrophages just replenish the intracellular lack of cholesterol mediated by statins- inhibition of biosynthesis by uptaking it from outside. However authors need to assess this by mimicking metabolic syndrome-like situations and the environment where macs are surrounded by physiologically/pathologically. (This is the reason why a mouse model of metabolic syndrome may help).
  7. It is not clear if authors want to conclude that ERK and NFkB are activated by protein prenylation, which is regulated by the statins. If so, authors need to show it better: they could investigate the inflammatory markers with Sim or Cer in the presence of FPP or GGPP.
  8. For the effect elicited by MVA, or the one elicited by FPP or GGPP, authors are hinting to the fact that these pathways are indeed working in the macrophages and expression levels of the enzymes in these pathways may be differentially regulated by the two statins. It would be interesting to understand the flux of these pathways into cholesterol as final product in the macrophages and what intermediates are preferentially synthetized under the different inflammatory stimuli. It would be helpful then to quantify all these intermediates in macrophages. And following up on this, authors should provide information and cite literature about prenylation. Is protein prenylation important for LPS or chronic LPS or IL4 stimulations?

Author Response

We thank the editorial board member for his constructive suggestion to include more information to “describe the pathological significance of their findings in response to comment 1”.

We have included respective explanations in the revised version of the manuscript.

This manuscript is a resubmission of an earlier submission. The following is a list of the peer review reports and author responses from that submission.

Round 1

Reviewer 1 Report

Rebecca Linnenberger et al express the idea of addressing the role of statins in macrophage behavior. They investigate the levels or pro and anti inflammatory markers using 2 different statins together with a relatively new compound, bempedoic acid, used for hypercholesterolemia. They find that statins but not bempedoic not may impact, likely enhancing, some inflammatory characteristics, only in a short endotoxin treatment. Although vaguely expressed, the idea of assessing inflammatory phenotypes of macrophages under statin treatment may have a reasonable impact not only for immunology but also for the whole field of cardiovascular diseases; however the manuscript presented does not sufficiently demonstrate substantial macrophage phenotype skewing that could be related or translated to in vivo pathology. The aim of the study itself appears not clear therefore it is hard to follow the conclusions that authors want to draw from each experiment performed. Moreover, the model undertaken for the authors’ study needs to be extensively revised. I am not in favor of publication of this manuscript in this journal.

major

  1. From the abstract it is not clear what the authors suggest as aim of the study. What would the investigation of inflammatory property of macrophages under statins treatment lead the authors to? What would be the conclusion in light of the microenvironment where macrophage are in situ in the situation of hypercholesterolemia or cardiovascular disease?
  2. It would be important to explain the molecular mechanisms at the basis for the beneficial effects of statins (antioxidant, anti-inflammatory effects, plaque stabilization). Moreover, authors should add more information from the study where it has been suggested that statins may activate inflammatory pathways.
  3. Authors should briefly summarize the data from the literature about conflicting ideas about the effects of statins on macrophages.
  4. It is not clear what is the active compound of bempedoic acid that acts on Acly. It appears to be involved in a pathway way upstream of the actual synthesis of cholesterol; authors should articulate a little about the main findings regarding this compound and other readouts like for example neutral lipids-triglycerides levels. Moreover, if bempedoic acid is a prodrug what is the evidence that could act on macrophages in a specific way? Authors should clarify the rationale for using this compound in their study.
  5. Authors should describe the muscular side effects of statins explaining the mechanisms.
  6. It would be important to write in the legend of each figure what statistical tests have been performed in the represented graphs.
  7. In paragraph 2.1, the part about M2 markers should be better explained as it is not clearly presented. Since the authors seem to point to a pro-inflammatory skewing of macrophages with statins, how do the authors explain that some M2 markers are increased? Authors should consider assessing more M2 markers, like Tgfb, Cd163 (Tsc22d3 needs to be better defined as well). Moreover, it would be interesting if authors could assess surface markers for both M1 and M2 phenotypes.
  8. It is not clear why phagocytic activity with statins +LPS is significantly different in figure 3D but not in 3G.
  9. It would be important that author explained why MVA effect is detectable only in the presence of cerivastatin but not of simvastatin. Could cerivastatin and simvastatin mediate pro inflammatory effect via different pathways?
  10. Authors should provide literature in support of treatment with simvastatin or cerivastatin beforehand for 24h. it is also not clear if, when M1/M2 stimuli are given, the cells are treated for a total of 48h (24h + 24h) or not. Is a change of extracellular medium included in this 48h protocol? What happens if these drugs are added afterthe LPS or M1/M2stimuli? Moreover it is not clear if simvastatin or cerivastatin plus mevalonate are added simultaneously.
  11. Authors should be more clear about the concentration of bempedoic acid used for their study. Please provide literature in support. What happens if they increased its concentration?
  12. It would be important that authors explained the rationale for investigation of a short term LPS-type of inflammatory response. It is hard to understand why authors pursue a short term inflammation model if the people that are treated with statins likely have chronic inflammation, insulin resistance and possibly atherosclerosis going on. Authors should pick a different model of activation, one that is more suited for the purpose on understanding what statins may do to macrophages. For example macrophages treated with oxLDL or other types of M1 and M2s (not the classical) may be considered. As for point n10, authors should maybe add statins at 48h of LPS treatment, or perform low dose LPS+IFNγ for longer times.
  13. Authors mention discrepancies in the literature based on the different methods investigated especially about in vivo ad ex vivo macrophage phagocytic activity. They should give more details about these studies. Moreover, authors should use for their model (see point 10 and 12) something more comparable to the in vivo settings. Like resident macrophages and blood recruited.
  14. Besides ERK, authors should assess the role of PKC in the phenotype they report.
  15. When mentioning bempedoic acid and its described anti-inflammatory activities, authors should be more specific about the prodrug effect vs the active compound. Moreover, following point n4, it follows that the rationale for the use of bempedoic cid in unclear. It is not well expressed if the aim of the authors is to demonstrate that bempedoic acid may be safer as macrophages may not be skewed in their phenotype as they do by the statins. Furthermore, it would be important that authors made clear what phenotype of macrophages may be important to achieve (or maintain) during hypercholesterolemia or cardiovascular diseases. What phenotype would be desired to ameliorate pathologic condition?

Minor:

  1. Authors should be more specific on what statins are currently approved for.
  2. All abbreviations must bel spelled entirely the first time they are mentioned.
  3. Title of 2.1 could be changed in “modulation” rather than “induction”.
  4. Authors should represent individual points in all their graphs (scattered plots) as the variability in their experiments seem to be high.
  5. In figure 1b please try to zoom a bit on the ctrls. The differences cannot be appreciated.
  6. Title of 2.2 needs to be re-written.
  7. Figure 3A should be enlarged (at the level of uM zoom of each panel)
  8. Authors should not refer to “data not shown” unless they decide to disclose additional information.
  9. In the methods, when authors describe BMM isolation and culture, it is not clear after the adherence overnight step, if they collect non adherent cells or they discard them for further macrophage culture.

Reviewer 2 Report

The authors found that statins (simvastatin, cerivastatin) may have an immunomodulatory effect on macrophage polarization. Moreover, the authors found that bempedoid acid, which is now a new drug to regulate blood cholesterol, is not affected by this activity.
The work is very interesting. Extensive results draw attention to the fact that the tested statins may also have a different activity than the commonly known one.

Author Response

Reviewer 2

 The authors found that statins (simvastatin, cerivastatin) may have an immunomodulatory effect on macrophage polarization. Moreover, the authors found that bempedoid acid, which is now a new drug to regulate blood cholesterol, is not affected by this activity.
The work is very interesting. Extensive results draw attention to the fact that the tested statins may also have a different activity than the commonly known one.

We would like to thank the reviewer for appreciating our comprehensive set of data and their interpretation.

Round 2

Reviewer 1 Report

Rebecca Linnenberger et al have provided a revised manuscript where they only partially address the reviewer’s comments. Although they clearly summarize the results of their investigation and explain some of the major inconsistencies, the manuscript does not rationally reflect the authors’ aim and the conclusions drawn are still not supported by valid experimentation.

Many previous comments still stand, in particular:

  • In the response to point n4, the argument is still not clear. The fact that authors mention a transporter in macrophages means very little if there is no literature cited that support the entrance of the prodrug in the cell via that transporter and no studies are mentioned about its metabolism.
  • In the response to point n9, is the speculation authors make (regarding kinetic profile of cerivastatin and simvastatin) valid also for bempedoic acid? If bempedoic acid blocks the same pathway how do they explain that they don’t see similar effect? Is the concentration utilized too low? (Variations in Erk indeed are very little). See next point
  • Regarding figS1, in Bmdm it looks like at 50uM cells are only partially and not substantially impaired in viability by bempedoic acid.
  • It is not clear why authors didn’t follow the same procedure of pretreating for 24h with statins before M1/M2 stimuli like they adopted for LPS alone.
  • A reference is needed for the method of collecting non adherent cells during Bmdm culture.